# The Cost-Effectiveness of the Human Papilloma Virus Vaccination in Asia Pacific Countries: What Lessons Can Indonesia Learn?—A Systematic Review

**DOI:** 10.3390/vaccines13060593

**Published:** 2025-05-30

**Authors:** Suzanna Patricia Mongan, Joshua Byrnes, Hansoo Kim

**Affiliations:** School of Medicine and Dentistry, Griffith University, Gold Coast, QLD 4222, Australia; j.byrnes@griffith.edu.au (J.B.); hansoo.kim@griffith.edu.au (H.K.)

**Keywords:** cervical cancer, human papillomavirus vaccination, cost-effectiveness, Asia Pacific, Indonesia

## Abstract

**Background/Objectives:** Cervical cancer is a significant issue in Indonesia, with many cases diagnosed at advanced stages. Although the human papillomavirus (HPV) vaccination has long been recommended by the WHO, it was only recently included in Indonesia’s immunization program in 2023. This study aimed to examine the existing prevention strategies and their effectiveness through systematic review of the existing literature. **Methods**: We searched for cost-effectiveness studies of HPV vaccination in Asia Pacific countries from inception until 23 July 2023, using MEDLINE, Embase, and the Cochrane Library databases. The search strategy included keywords and subject terms for primary prevention, HPV vaccination, cervical cancer, and selected Asia Pacific Countries (Thailand, Vietnam, China, Singapore, Malaysia, Philippines, Korea, Japan, Taiwan, Australia, New Zealand, and Hong Kong). Studies selected were limited to original research articles with full text published in English in peer-reviewed journals, describing the cost-effectiveness of HPV vaccination in cervical cancer prevention in the Asia Pacific countries. Studies were excluded if there were no full text was available, if it was the wrong study design, non-English, or not based in the specific Asia Pacific countries selected. The titles and abstracts were screened, followed by full-text reviews using Covidence software, and analyzed using Excel. **Results**: Forty-three studies were included for review: 51% in high-income countries (HICs), 37% in upper-middle-income countries (UMICs), and 12% in low-middle-income countries (LMICs). All studies concluded that HPV vaccination is more cost-effective than screening alone. Nonavalent HPV vaccines were more cost-effective in HICs (80%), bivalent vaccines were more cost-effective in UMICs (66%), and gender-neutral vaccination was cost-effective compared to screening in all studies conducted. **Conclusions**: HPV vaccination is a cost-effective prevention strategy for cervical cancer across all resource settings, offering greater value compared to screening alone. Selecting the most economically viable vaccine type and expanding to gender-neutral vaccination could enhance early prevention efforts. These findings offer guidance for Indonesia in designing evidence-based HPV vaccination policies as a part of national cancer control efforts. Further investigation is necessary to determine the optimal strategy for HPV vaccination in Indonesia.

## 1. Introduction

Cervical cancer is still a burden worldwide. It ranks as the fourth most frequently diagnosed cancer and cause of cancer death in women globally, with 604,127 new cases in 2020. In low- and middle-income countries, cervical cancer contributed to approximately 90% of the 342,000 cases [1,2]. It has the highest incidence rates and is the leading cause of cancer mortality in many African, Melanesian, South American, and Southeast Asia countries [1]. In 2020, cervical cancer caused an estimated 190,874 new cases and 116,015 deaths in the World Health Organization (WHO) Southeast Asia region, with the global age-standardized rate of 13.4% [2,3]. Although this region accounts for only 8.6% of the global population, it contributed to approximately 32% of cervical cancer incidents and 34% of cervical cancer deaths worldwide [3,4].

Indonesia has the highest incidence rate of cervical cancer in the Southeast Asia region, and the second highest in Asia [5]. The number of new cases of cervical cancer in Indonesia, according to GLOBOCAN, in 2022 is 36,964 cases, and it is the second most frequent cancer in females after breast cancer [6]. The age-standardized cervical cancer incidence and mortality rates, respectively, are as high as 23.4 and 13.9 per 100,000 in Indonesia [3]. It is projected to continue to increase as patients often present with advanced-stage cancer upon diagnosis.

Persistent infection of high-risk human papillomavirus (HPV) is known to cause about 99.7% of cervical cancer cases [7,8]. Two of the most oncogenic HPV, HPV 16 and 18, are responsible for 71% of cervical cancer reported globally. Most sexually active women and men may become infected with HPV in their lifetime. In total, 90% of the infection will regress, but a small proportion (approximately 10%) of infections can persist for several years and progress to precancerous lesions. If it is left untreated, it can progress to invasive cancer [3,7,8,9].

Since 2020, the WHO has established cervical cancer as a public health problem globally and has set a global strategy to accelerate the elimination of cervical cancer by the end of the century. To achieve this by 2030, HPV vaccination should be included in all national immunization programs to ensure that at least 90% of girls are fully vaccinated by age 15; 70% of women have been screened using a high-performance test by 35 years of age and again by 45 years of age; and 90% of women diagnosed with cervical disease received necessary treatment. This approach is expected to reduce 30% of cervical cancer deaths by 2030 [10,11].

Currently, all HPV vaccines have protection against the two most oncogenic HPV, type 16 and 18, which are the cause of 71% of cervical cancer. The bivalent and quadrivalent vaccines also offer some additional protection against HPV types 31, 33, and 45, which are associated with 13% of cervical cancers, for total protection against high-risk HPV types linked to 84% of cervical cancers. The nonavalent vaccine, on the other hand, provides protection against five additional high-risk HPV types (31, 33, 45, 52, and 58), which are associated with 18% of all cervical cancers, thereby protecting against 90% of high-risk HPV types known to cause cervical cancer [3,11].

Studies have shown that implementing an HPV vaccination program for girls can significantly decrease the incidence rate of cervical cancer in low- and middle-income countries (LMICs). There is a projected reduction of 89.4%, decreasing from 19.8 to 2.1 cases per 100,000 women over the next century. This has the potential to prevent 61 million cases of cervical cancer during this period [3,12].

Globally, 125 countries have introduced the HPV vaccine in their National Immunization Program (NIP) for girls and 47 countries for boys [11,13]. Australia was among the first countries to implement HPV vaccination as its national immunization program. In 2020, 80.5% of girls in Australia were fully vaccinated by age 15, making the global targets within its reach [14]. Despite having a high incidence of cervical cancer, the overall HPV vaccination coverage in Indonesia is still low [5,15]. Indonesia has just included the HPV vaccination program in their NIP in mid-2023, following the subnational introduction in the province of Jakarta in 2016, in Yogyakarta province, partly in 2017 and in all districts in 2019, in East Java province in 2017, and in Central Java, South Sulawesi, and the North Sulawesi provinces in 2018 [3,16].

One major challenge to vaccination uptake, particularly in low- and middle-income countries, is the combination of public awareness, inadequate health infrastructure, socioeconomic disparities, and fragmented healthcare systems, all of which restrict access and reduce coverage [17]. These structural barriers are further complicated by low community knowledge and behavioral factors. A recent study among urban communities in Indonesia revealed that while attitudes toward HPV vaccination were generally positive, knowledge levels were low and vaccine uptake was poor, despite high self-reported willingness to be vaccinated. Sociodemographic characteristics, such as education level and gender, significantly influenced the knowledge, attitude, and practice levels [18]. These findings underscore the importance of public education, gender-sensitive strategies, and targeted communication efforts with policy and infrastructure development to achieve high vaccine coverage.

The primary objective of this study is to evaluate the cost-effectiveness of HPV vaccination for cervical cancer prevention across Asia Pacific countries. The secondary objectives are to compare the cost-effectiveness of different types of HPV vaccines, assess the value of gender-neutral vaccination approaches, and identify strategies most applicable to Indonesia. Research questions included the following:What types of HPV vaccines are most cost-effective across different economic settings?Is gender-neutral vaccination consistently found to be cost-effective across regions?

By evaluating the cost-effectiveness of HPV vaccination strategies in the Asia Pacific region, this study aims to assist policymakers in Indonesia in implementing the most efficient HPV vaccination strategies within the broader context of national cancer prevention programs.

## 2. Materials and Methods

### 2.1. Study Design

This systematic review was conducted to identify and evaluate the existing literature on the cost-effectiveness of HPV vaccination in the Asia Pacific countries. This review followed the Preferred Reporting Items for Systematic Reviews and Meta-Analyses (PRISMA) 2020 guidelines [19]. A completed PRISMA checklist is provided in Appendix A. This review protocol was not registered.

### 2.2. Eligibility Criteria

This review included original, peer-reviewed research articles published in English with full-text availability that specifically evaluated the cost-effectiveness of HPV vaccination in cervical cancer prevention in twelve Asia Pacific countries (Thailand, Vietnam, China, Singapore, Malaysia, Philippines, Korea, Japan, Taiwan, Australia, New Zealand, and Hong Kong). Studies were excluded if they were not economic evaluations; were conducted outside the specified countries; had inappropriate settings, populations, indications, interventions, study designs, or outcomes; lacked full-text availability or were not written in English.

### 2.3. Data Sources and Search Strategy

A comprehensive and systematic search was conducted across Embase, Medline (via OVID), and Cochrane databases from inception to 3 July 2023, with an updated search on 23 July 2023. The search includes a combination of MeSH terms and Boolean operations with AND/OR using keywords such as “primary prevention”, “HPV vaccination”, “cervical cancer”, as well as specific keywords related to selected Asia Pacific countries such as ‘Thailand”, “Vietnam”, “China”, “Singapore”, “Malaysia”, “Philippines”, “Korea”, “Japan”, “Taiwan”, “Australia”, “New Zealand”, and “Hong Kong”. The detailed search terms are listed in Appendix A.

Countries were classified according to the World Bank income classification at the time of the study [20]. Based on this, the included countries were categorized as high-income countries (HICs), upper-middle-income-countries (UMICs), or lower-middle-income countries (LMICs).

### 2.4. Data Selection

All search results from the three databases, including those obtained through citation searching, were exported into Covidence Systematic Review Software, Veritas Health Innovation, Melbourne, Australia (Available at www.covidence.org). Duplicates were automatically removed using Covidence, and an additional nine duplicates were identified and removed manually. Initial titles and abstracts were screened, resulting in the exclusion of irrelevant articles. The remaining articles were assessed for eligibility based on the inclusion criteria. The final selected articles underwent data extraction and analysis. The data selection involved two reviewers. Disagreements were resolved through discussion and consensus or by a third reviewer. The selection process is summarized in the PRISMA flow diagram (Figure 1).

### 2.5. Data Collection Process

A comprehensive data matrix was created in Microsoft Excel to summarize the key characteristics of the studies included in the research, extracting information such as authorship, study settings, perspectives, thresholds, outcome-related parameters, and other essential data points. The results of the data extraction are presented in Appendix A.

### 2.6. Quality Assessment

The Consolidated Health Economic Evaluation Reporting Standards (CHEERS) 2022 checklist was used to assess the quality of reporting of each health economic study included in this review [21]. See Appendix A.

### 2.7. Data Synthesis

Due to the heterogeneity of models, settings, and cost structures, a meta-analysis was not feasible. We conducted a narrative synthesis and comparative analysis across countries and model characteristics. Results are presented in Appendix A.

## 3. Results

Five thousand five hundred and twenty-five articles were yielded through the initial search from databases. Nine studies were found by citation searching. Two thousand two hundred and seventy-four articles were removed after deduplication including the nine studies that were found from citation searching. Six hundred and ninety-one studies were accessed for eligibility. Forty-three studies were included after excluding studies that did not meet the inclusion criteria. The reasons for exclusion are shown in Figure 1.

This study includes forty-three health-economics studies on HPV vaccination from eleven countries in the Asia Pacific region. The countries included are China (n = 10) [22,23,24,25,26,27,28,29,30,31], Singapore (n = 5) [32,33,34,35,36], Japan (n = 5) [37,38,39,40,41], Taiwan (n = 5) [42,43,44,45,46], Malaysia (n = 4) [47,48,49,50], Australia (n = 3) [51,52,53], Vietnam (n = 3) [54,55,56], Thailand (n = 3) [57,58,59], Hong Kong (n = 2) [60,61], Philippines (n = 2) [62,63], and New Zealand (n = 1) [64]. None of the studies were conducted in Korea. The distribution of studies across income categories, according to the World Bank [20], is as follows: high-income countries (HICs) had the highest percentage of studies (51%), followed by upper-middle-income countries (UMICs) with 37%, and lower middle-income Countries (LMICs) with a contribution of 12%. Most studies (50%) were conducted without a conflict of interest. The detailed overview of the study distribution is shown in Table 1.

Of all the studies, 16 (37%) were funded by pharmaceutical companies that manufacture the HPV vaccine, such as Merck, Sharp & Dohme or GlaxoSmithKline [24,29,32,35,40,42,44,46,47,49,50,59,60,61,62]. Nine studies (21%) were funded by national research funds [26,27,30,31,38,45,53,64], four (9%) were supported by the Bill & Melinda Gates Foundation [23,54,55,57], and three (7%) received funding from the educational sector [41,52,63]. One study (2%), conducted in a low-middle-income country, received funding from WHO and GAVI [56]. Only one study (2%) did not obtain any funding for its research [28]. The remaining nine studies (21%) did not state any funding resources [22,33,34,36,37,39,43,48,58]. Notably, all studies conducted in low-middle-income countries received external funding for their research on the HPV vaccine programs. More than half of the studies (53%) declared no conflict of interest [22,23,25,26,27,28,30,31,36,38,40,41,45,47,48,52,55,56,57,58,60,61,64], 40% declared a conflict of interest [24,29,32,33,34,35,37,42,43,44,46,49,51,53,59,62,63], and 7% did not disclose any in their study [39,50,54]. See Table 1.

In this systematic review, most studies (41.8%) focused on comparing the cost-effectiveness of HPV vaccination alone or combined with screening, compared to no vaccination or screening only [23,24,26,29,31,37,38,39,40,41,43,45,48,50,51,54,58,59]. Additionally, one of those studies compared the cost-effectiveness of different types of vaccines [50]. Fifteen studies (34.9%) specifically compared the cost-effectiveness of various types of HPV vaccines currently available [22,27,28,30,32,33,35,42,44,46,49,53,56,62,63]. Five studies (11.6%) explored the cost-effectiveness of different vaccination strategies and delivery methods [34,52,57,60,64]. Three studies (7%) investigated the cost-effectiveness of gender-neutral vaccination compared to female-only vaccination [36,55,61]. In comparison, two studies (4.7%) examined the cost-effectiveness of different numbers of HPV vaccine doses [25,47]. See Figure 2.

As shown in Figure 2, out of the 43 studies included in the review, 19 (44.2%) used Markov models in their HPV cost-effectiveness studies [24,25,27,29,32,35,38,39,41,44,45,46,47,49,51,58,62,64]. Eleven studies (25.6%) utilized dynamic economic models [26,33,34,40,42,43,53,55,59,60,61], five studies (11.6%) used PRIME models [22,36,52,56,65], three studies (7%) used Monte Carlo simulation models [23,54,57], one study (2.3%) used Markov Chain Monte Carlo models [28], and one (2.3%) used continuous time models [37]. One (2.3%) used UNIVAC models in their health economic analysis [63]. However, two studies (4.7%) did not specify the model used in their research [48,50]. Sensitivity analysis was performed in all studies (86%) except for six (14%) that did not mention it in their publication [23,34,51,55,57,59].

Regarding cost-effectiveness, the studies included in the review varied in their perspective. Thirty studies (69.8%) used a health system perspective, while seven (16.3%) adopted a societal perspective [22,27,40,41,54,55,57]. Five studies (11.6%) utilized both the health system and societal perspectives [39,48,50,52,63], and one study (2.3%) did not disclose the perspective used [43]. Moreover, most studies (74.3%) utilized the threshold recommended by the WHO, which is based on the national gross domestic product (GDP) or three times the GDP per capita. However, seven studies (16.3%) were based on the willingness-to-pay (WTP) threshold [28,31,36,37,39,41,53]. Two studies (4.7%) used GDP and WTP thresholds [27,46], while two (4.7%) did not specify the threshold utilized [47,51].

Most studies (81.4%) used quality-adjusted life years (QALYs) as the cost-effectiveness unit. Five studies (11.6%) used disability-adjusted life years (DALYs) [26,30,52,56,63], and seven studies (7%) used years of life saved (YLS) [23,54,57].

Regarding vaccine coverage, most studies (93%) have set the coverage rate at 70% or above. However, one study (2.3%) by Konno et al. used a coverage rate of 50% [39], while two other studies (4.7%) did not disclose their coverage rate [33,46]. Moreover, 17 studies (40%) had assumed a vaccination coverage rate of 90% or more, which aligns with the target set by the World Health Organization [10,11]. Three studies (7%) considered vaccine coverage rates ranging from 0% to 90% [26,43,55]. Additionally, certain studies (14%) varied the vaccine coverage rate based on the type of vaccines, populations (e.g., females or males), number of doses, and population age [27,34,53,60,63,64].

As for vaccine efficacy, most studies (60%) considered vaccine efficacy rates ranging from 80 to 100%. The vaccine efficacy in seven studies (16%) depended on the types of vaccines, the number of doses received, and the population, whether they were females or males [27,33,34,46,53,55,63]. Furthermore, in the context of vaccine protection, 88.4% of the studies assumed lifetime vaccine protection, except for two studies (4.7%) with a shorter duration of 20 years of protection [52,64], and three studies (7%) did not state any information in their publication [26,30,56]. More than half of the studies (60.5%) did not consider the indirect effect due to vaccination, but eleven of the studies (25.6%) specified herd immunity due to vaccination [28,37,42,51,52,53,55,59,60,61,64], and six studies (14%) did not state any information [33,34,39,40,41,54].

All studies indicate that HPV vaccination is a cost-effective intervention for preventing cervical cancer. However, the cost-effectiveness varies among different types of vaccines, age groups, and geographical settings.

Various studies have consistently found that HPV vaccination is a more cost-effective prevention strategy for cervical cancer compared to no vaccination or screening-only scenarios [23,24,25,26,29,30,31,32,38,39,40,41,43,45,46,48,50,51,54,56,57,58,59,63]. Any type of HPV vaccination as an addition to screening is a more cost-effective intervention to prevent cervical cancer compared to screening only, in all studies [23,26,29,31,32,38,39,41,48,50,51,54]. Catch-up vaccination is also considered cost-effective in six studies when added to pre-adolescent female vaccination [37,43,45,59,60]. However, several studies conducted in low and middle-income countries have highlighted that vaccination at a lower cost or significant price reduction in vaccines is necessary to achieve a cost-effective threshold [26,29,31,54,56,57]. Ma et al. suggested that scaling up cervical cancer screening in adult women is the most affordable strategy for maximal health benefits unless there is a substantial price reduction to conduct a universal HPV vaccination program. Domestic production of vaccines at a lower cost can be an alternative solution [26].

In the context of types of HPV vaccines, the result differs in various studies conducted. Specifically, three studies found the quadrivalent vaccine (4vHPV) more cost-effective than the bivalent vaccine (2vHPV) [27,32,50]. However, five other studies found the bivalent vaccine were more cost-effective over the quadrivalent vaccine [35,44,46,49,62]. As for the nonavalent vaccine (9vHPV), three studies found it more cost-effective than the bivalent vaccine [27,34,42], while three other studies found the opposite [22,28,46]. Comparing nonavalent with quadrivalent, four studies found nonavalent to be more cost-effective [33,35,37,53]. However, one study reported the opposite result, with the quadrivalent vaccine being more cost-effective over the nonavalent vaccine [22]. In a separate study, Phua et al. found that the nonavalent vaccine was not a cost-effective option due to its high price point, and significant price reductions would be necessary to make it a viable choice [28]. Similarly, Jiang et al. arrived at the same conclusion that nonavalent vaccine is not cost-effective when compared to both quadrivalent and bivalent vaccines, and is only slightly more cost-effective compared to not getting vaccinated at all, given its current price [22]. See Figure 3.

Three studies were included in the review that evaluated the cost-effectiveness of implementing gender-neutral vaccination compared to female-only vaccination. Cheung et al. reported that gender-neutral vaccination using the nonavalent vaccine was more cost-effective than female-only vaccination [61]. However, the study conducted by Wahab et al. found gender-neutral vaccination to be not cost-effective when the nonavalent vaccine is used at its current price. Still, it can be cost-effective when the bivalent vaccine is used [36]. Meanwhile, Sharma et al. concluded in their study conducted in a low-resource setting in Vietnam that gender-neutral vaccination using quadrivalent vaccines is cost-effective compared to female-only vaccination at low cost but with little benefit. They suggested that emphasizing high coverage of HPV vaccination of girls could be a more efficient approach [55].

## 4. Discussion

HPV vaccination is one of the essential cornerstones in cervical cancer prevention worldwide. The introduction of national HPV vaccination programs is subject to various factors, such as the burden of cervical cancer, the effectiveness of existing prevention methods, including the type of cervical screening utilized, and national financial resources [66]. Cost-effectiveness studies offer policymakers a valuable tool to assess whether including HPV vaccination alongside cervical screening would yield more significant health and economic advantages within reasonable budget constraints. Indonesia, despite being categorized as an upper-middle-income country, continues to face a significant burden of cervical cancer. One of the contributing factors to this is the low coverage rate for cervical cancer screening. However, the nation has taken a step forward by recently introducing HPV vaccination into its national immunization program starting mid-2023 [3,16].

It is essential to examine the available evidence and identify key lessons that Indonesia can learn from to optimize the effectiveness of its cervical cancer prevention. We explored the cost-effectiveness of HPV vaccination in 11 Asia Pacific nations, utilizing 43 cost-effectiveness evaluations as a reference point. This valuable material can serve as a benchmark for future studies on strategies and interventions aimed at preventing cervical cancer in Indonesia, including HPV vaccination. Most studies were conducted in high-income countries, with five studies performed in LMICs [28,32,33,34,35,36,37,38,39,40,41,42,43,44,45,46,51,52,53,55,56,60,61,63,64].

Using different models (such as dynamic, static, or hybrid) and varying assumptions about the decision problem (such as vaccine price, comparator, perspective, outcomes included vaccine coverage, and decision-making threshold) in cost-effectiveness studies can make it difficult to compare study results directly [67]. However, across the studies reviewed, HPV vaccination is consistently more cost-effective than no vaccination or screening alone. HPV vaccination is also the most cost-effective strategy when screening coverage is low, and the feasibility and long-term screening adherence are uncertain. Combining any HPV vaccination with screening proves to be a more cost-effective approach than screening alone. This aligns with the cost-utility analysis of HPV vaccination in Indonesia, conducted by Setiawan et al., which concluded that HPV vaccination is a cost-effective strategy for 12-year-old females in addition to visual inspection with acetic acid (VIA) screening in Indonesia [66].

The cost-effectiveness of different types of HPV vaccines remains a subject of variability and disagreement across various studies in this review. The comparison between quadrivalent (4vHPV), bivalent (2vHPV), and nonavalent (9vHPV) vaccines has yielded different findings. These conflicting results underscore the complexity of determining the most cost-effective HPV vaccine type. Factors such as vaccine prices, effectiveness, and the specific population context can significantly impact the outcomes. A study by Setiawan et al. that included all three types of vaccines in their cost-effectiveness model in Indonesia concluded that all three vaccines are cost-effective, with the highest reduction in cervical cancer incidence and mortality generated, but the nonavalent vaccine and bivalent vaccine had the lowest ICER [68]. Policymakers and public health authorities should consider these nuanced findings when implementing HPV vaccination programs, weighing the potential benefits against the associated costs and vaccine types.

The studies on gender-neutral vaccination in this review highlight the challenges in determining the cost-effectiveness of gender-neutral vaccination compared to female-only vaccination. The findings of these studies underscore the importance of examining the cost-effectiveness of gender-neutral versus female-only vaccination in the context of specific countries and vaccine types. Further research is needed to determine the most optimal vaccination strategies in different settings.

The cost and type of vaccine, as well as the specific context, all play a significant role in determining the economic feasibility of these strategies. Policymakers and public health officials should consider these findings when developing vaccination policies, emphasizing the importance of tailored approaches that consider local conditions, vaccine characteristics, and cost considerations. However, it is worth noting that despite the significance of this issue, there is a concerning lack of cost-effectiveness studies on gender-neutral HPV vaccination in Indonesia, and further research is needed to address this knowledge gap.

Limited research has been conducted on the cost-effectiveness of HPV vaccination and cervical screening in Indonesia. Therefore, it is essential to perform further studies to determine the cost-effectiveness of HPV vaccination with different types of vaccines, gender-neutral vaccination, catch-up vaccination, and the combination of HPV vaccination with screenings such as VIA, pap smears, and the newly recommended HPV DNA screening by the WHO to achieve higher coverage. These studies should be conducted from both a healthcare and societal perspective, using real-life data on direct costs, indirect costs, and health resources used. Such studies can help policymakers and healthcare professionals make informed decisions about the most effective and efficient strategies for HPV prevention and screening in Indonesia.

### 4.1. Perspective for Clinical Practice

The findings of this review offer several important implications for clinical practice in Indonesia. HPV vaccination, particularly when integrated with cervical cancer screening, is a consistently cost-effective strategy and should be prioritized as a part of a comprehensive cervical cancer prevention program [20,23,26,28,29,35,36,38,45,47,48,51]. The recent nationwide rollout of HPV vaccination for girls is a promising advancement, but further efforts are necessary to improve uptake and coverage. Strengthening the delivery of HPV vaccines is critical. School-based vaccination programs and strong community engagement are essential strategies to enhance coverage and ensure equity across diverse populations [47,60,64]. Clinicians and healthcare providers play a pivotal role in this process, not only as implementers but also as advocates, educators, and system-level facilitators. Addressing vaccine hesitancy, misinformation, and health system fragmentation remains a key clinical priority. Importantly, enhancing the knowledge and positive attitudes of the community towards the cervical cancer prevention program has been shown to significantly influence the acceptance of HPV vaccination and screenings in the community [18].

Health professionals must also continue to underscore the importance of cervical cancer screening in addition to HPV vaccination. Healthcare systems should aim to vaccinate and establish coordinated screening pathways accessible through primary care. Future screening policies in Indonesia should integrate more sensitive methods like HPV DNA testing in accordance with the WHO recommendations [69].

Furthermore, vaccine choice and affordability considerations must be guided by cost-effectiveness considerations. Policymakers may benefit from adopting tendering processes that prioritize vaccine types based on negotiated pricing, expected health gains, and national budget constraints. Although current efforts focus on female-only vaccination, evidence from other countries suggests that gender-neutral strategies can be cost-effective, particularly when female coverage is suboptimal or HPV transmission among males remains high [36,61,70]. Modeling studies and pilot programs could provide valuable insights into the feasibility and economic value of such expanded strategies in Indonesia.

Moreover, clinicians and researchers in Indonesia should work collaboratively to generate local evidence on vaccine effectiveness, cost, and societal preferences. Active clinician involvement in implementation studies and impact evaluations will support more robust, evidence-based policymaking.

The successful integration of HPV vaccination into clinical and public health practice in Indonesia will depend on a multifaceted approach that combines scientific evidence with contextual relevance. By addressing delivery, equity, and strategic integration with screening, clinicians and public health professionals can help ensure that HPV vaccination efforts are not only scientifically justified but also socially and operationally effective.

### 4.2. Strengths and Limitations

This systematic review provides important insights into potential next steps for Indonesia in its approach to cervical cancer prevention. However, it is not without limitations. Most included studies were conducted in high-income countries, which may limit the generalizability of findings to the Indonesian context. Differences in healthcare infrastructure, vaccination costs, and population behaviors could influence real-world outcomes. The restriction to studies published in English may have excluded relevant research available in local languages, which could be particularly significant in countries with economic profiles similar to that of Indonesia. Additionally, the review only incorporated studies from a limited number of Asia Pacific countries, thereby overlooking relevant research from other nations that share similar characteristics to Indonesia. Finally, variability in modeling, assumptions, comparators, and outcome measures across studies limits the ability to directly compare findings or draw uniform conclusions.

## 5. Conclusions

This systematic review demonstrates that HPV vaccination is generally a cost-effective strategy for cervical cancer prevention across various Asia Pacific countries. Despite differences in modeling approaches, assumptions, and healthcare contexts, HPV vaccination consistently provides greater health and economic benefits compared to no vaccination or screening alone. These findings reinforce the importance of implementing HPV vaccination as a part of a comprehensive prevention strategy, particularly in settings with low screening coverage and limited resources. For Indonesia, these insights can guide the development of evidence-based policies tailored to local conditions. Further context-specific research is necessary to evaluate the cost-effectiveness of different vaccine types, gender-neutral strategies, catch-up vaccination, and the integration of vaccination with various cervical screening methods to support the long-term sustainability and effectiveness of the Indonesian national cervical cancer prevention efforts.

## Figures and Tables

**Figure 1 vaccines-13-00593-f001:**
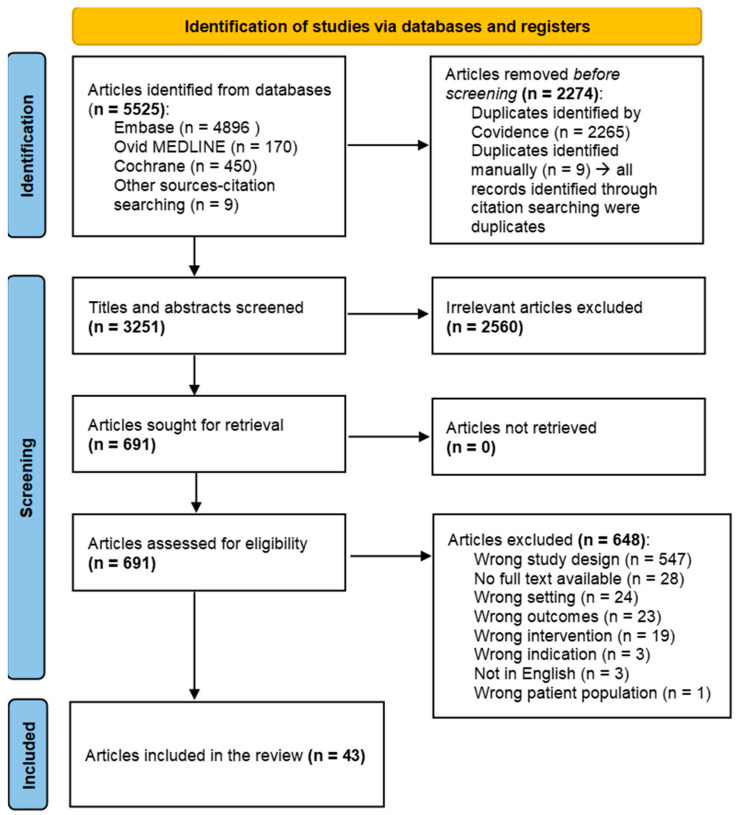
PRISMA 2020 flow diagram showing the study selection process. Note: All citation searching records (n = 9) were duplicates and removed.

**Figure 2 vaccines-13-00593-f002:**
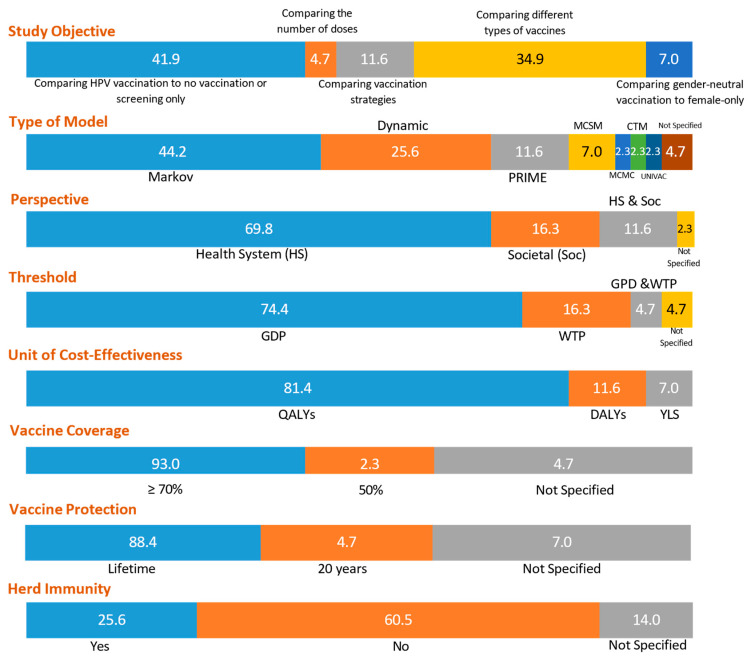
The characteristics of cost-effectiveness of HPV studies in percentage.

**Figure 3 vaccines-13-00593-f003:**
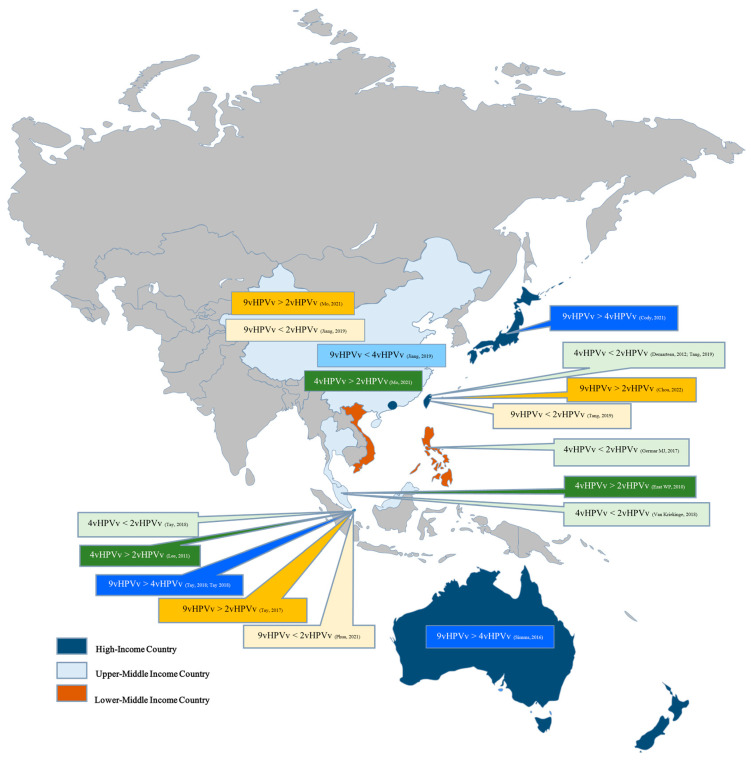
The cost-effectiveness of different types of HPV vaccines [16,21,22,26,27,28,29,31,36,38,40,43,44,47,56]. 9vHPVv: nonavalent HPV vaccine; 4v2HPV: quadrivalent HPV vaccine, 2vHPVv: bivalent HPV vaccine.

**Table 1 vaccines-13-00593-t001:** Characteristics of the forty-three included studies.

Characteristics	Number of Studies (n)	Percentage (%)
Selected articles	43	100
**Study setting**		
China	9	21
Singapore	6	14
Japan	5	12
Taiwan	5	12
Malaysia	4	9
Australia	3	7
Vietnam	3	7
Thailand	3	7
Hong Kong	2	5
Philippines	2	5
New Zealand	1	2
**Economic classification**		
High income	22	51
Upper-middle income	16	37
Lower-middle income	5	12
**The primary location of the first author**		
Research Institute	9	21
Research Group	8	19
Hospital or university	26	60
**Study Funder**		
Pharmaceutical company	16	37
National Fund	9	21
Bill & Melinda Gates Foundation	4	9
Education Fund	3	7
WHO and GAVI	1	2
Did not receive funding	1	2
Did not declare	9	21
**Conflict of interest**		
Yes	17	40
No	23	53
Not stated	3	7

WHO: World Health Organization; economic classification according to the World Bank Data 2023 [20].

## Data Availability

No new data were generated in this study. The data supporting the findings of this study are available in the Appendix A of this article.

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
