# Peer review of "The Cost-Effectiveness of the Human Papilloma Virus Vaccination in Asia Pacific Countries: What Lessons Can Indonesia Learn?—A Systematic Review"

_vaccines, 2025, doi:10.3390/vaccines13060593_

Round 1
Reviewer 1 Report
Comments and Suggestions for Authors
Dear Authors,
the comments in the annex file.
Best

Author Response
We sincerely thank you for the detailed and constructive feedback. Your helpful suggestions have greatly improved the structure, clarity, and overall quality of our manuscript.
Please see the attachment.

Reviewer 2 Report
Comments and Suggestions for Authors
Dear authors,
Your systematic review tackled an important topic regarding HPV vaccination and its cost-effectiveness. The paper was done according to PRISMA guidelines, but the PROSPERO number is missing. The introduction provides enough information regarding the topic; some letters are missing in some words (line 48, I think it is breast and not best; line 40, after 342,000, a word is missing, it seems the sentence is not finished). The material, method, and result parts are all okay. The data in the Figures is nicely presented. Discussion follows the results, limitations should be mentioned before the conclusion as a separate paragraph, and it seems like the final Conclusion/s are missing, the one that makes the point of the whole paper.
Best regards,
Author Response
Dear reviewer,
We sincerely appreciate your thoughtful comments and suggestions. Your input has been very helpful in enhancing the clarity and overall quality of our manuscript.
Please see the attachment.

Reviewer 3 Report
Comments and Suggestions for Authors
Dear authors, let me first congratulate you on a tremendous effort you did for this article. It is of high importance in todays healthcare system and in communications with the policy makers and all the relevant stakeholders to include the economic assessments.
However, before publication, there are few corrections to be made to the article:
Overall editing comment, please use the space before citing the reference.
ABSTRACT
The aim is stated:
‘This study aims to aid decision-makers in implementing the most effective prevention strategies in Indonesia’, is not well-formulated. Try ‘This study aimed to examine the existing prevention strategies and their effectiveness through systematic review of the existing literature.’
Introduction.
Line 39- put the reference number here as well at the end of the sentence
Line 42: Add some incidence rates
Line 43: Comparing to what % of the world population that this region presents?
Methods
As there are the results on the number of studies conducted in high and middle income countries, add the classification in the methods section. It is easier than for readers to follow. (after lines 115-117)
Results:
What would more dominant mean in this context? More frequently used? (lines 243-256)
Author Response
Dear reviewer,
Thank you for the constructive feedback and supportive remarks. Your feedback helped us improve the manuscript.
Please see the attachment.

Round 2
Reviewer 1 Report
Comments and Suggestions for Authors
Dear Authors,
very good job. Ready for publication.
Best.